# DynamicTree: Interactive Real Tree Animation via Sparse Voxel Spectrum

## Abstract

Generating dynamic and interactive 3D objects, such as trees, has wide applications in virtual reality, games, and world simulation. Nevertheless, existing methods still face various challenges in generating realistic 4D motion for complex real trees. In this paper, we propose *DynamicTree*, the first framework that can generate long-term, interactive animation of 3D Gaussian Splatting trees. Unlike prior optimization-based methods, our approach generates dynamics in a fast feed-forward manner. The key success of our approach is the use of a compact *sparse voxel spectrum* to represent the tree movement. Given a 3D tree from Gaussian Splatting reconstruction, our pipeline first generates mesh motion using the sparse voxel spectrum and then binds Gaussians to deform the mesh. Additionally, the proposed sparse voxel spectrum can also serve as a basis for fast modal analysis under external forces, allowing real-time interactive responses. To train our model, we also introduce *4DTree*, the first large-scale synthetic 4D tree dataset containing 8,786 animated tree meshes with semantic labels and 100-frame motion sequences. Extensive experiments demonstrate that our method achieves realistic and responsive tree animations, significantly outperforming existing approaches in both visual quality and computational efficiency. https://anonymous.4open.science/w/dynamictree-anonymous/

## 1 Introduction

With recent advances of neural radiance fields (NeRF) Mildenhall et al. (2021) and 3D Gaussian Splatting (3DGS) Kerbl et al. (2023), high-quality reconstruction and real-time rendering become feasible. Driven by these, there is high demand to make static reconstructions interactable, for immersive experiences like 3D games, movies, and virtual reality Jiang et al. (2024); Franke et al. (2025); Schieber et al. (2025). As a vital component of natural landscapes, tree animation can significantly enrich immersive digital experiences. For example, when viewing a reconstructed backyard on a VR headset, if trees can sway gently in the wind or respond to dragging interaction, it would significantly enhance immersion.

However, animating a realistic 3D tree remains challenging. Traditional tree animation methods Quigley et al. (2017); Pirk et al. (2017; 2014) typically construct physical tree models and then perform dynamic simulations. Although such approaches can generate realistic motion dynamics, they are either confined to synthetic tree models or require labor-intensive mesh refinements for high-quality rendering. Consequently, they are ill-suited for animating reconstructed 3DGS representations to faithfully reproduce real-world experiences.

To animate 3DGS trees, existing methods can be broadly categorized into 4D generation and physics-based simulation. 4D generation approaches first create a static 3DGS model from text or images Bahmani et al. (2024); Singer et al. (2023); Ling et al. (2024); Zheng et al. (2024); Xu et al. (2024); Miao et al. (2024); Yin et al. (2023), then use video diffusion models (VDMs) to optimize the 4D representation. These methods, however, involve costly per-scene optimization and often yield short sequences with poor geometric consistency. In contrast, physics-based approaches, such as PhysGaussian Xie et al. (2024), couple 3DGS with physical simulation engines like Material Point Method (MPM) Jiang et al. (2015), achieving better 3D consistency and longer motion sequences. Nonetheless, they rely on simplified assumptions (e.g., uniform material), which reduce realism and are computationally expensive, making them unsuitable for real-time applications.

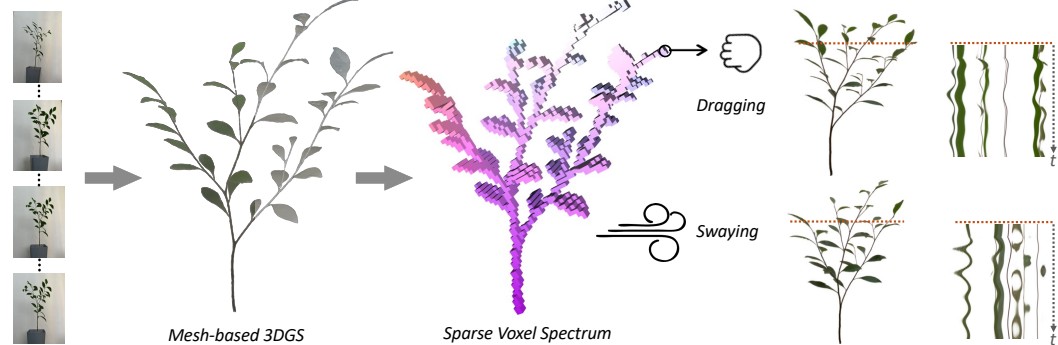

Figure 1: *DynamicTree* achieves long-term, realistic animation and dynamic interaction with real reconstructed 3DGS trees, which first generates mesh motion via a compact sparse voxel spectrum representation and then deforms the surface-bound Gaussian primitives. We visualize the slice of the generated motion at the orange scanline along the time dimension.

In this work, we introduce *DynamicTree*, a novel 4D generative framework for animating 3DGS trees with real-time interactions. Unlike VDM-based 4D generation methods, *DynamicTree* directly learns tree motion priors in 3D space, avoiding geometric inconsistency. Moreover, compared to computationally expensive physics-based optimization, our generative model directly synthesizes 3D tree motion in a feed-forward manner, achieving more than a hundred times acceleration. To train the model, we construct the first large-scale 4D tree motion dataset with $8,786$ animated meshes, semantic labels, and 100-frame sequences, generated via hierarchical branching simulation Weber & Penn (1995).

Still, even with this dataset, direct motion prediction in 3D space is challenging, as it requires both an efficient and robust representation of 3D motion. Since a reconstructed 3DGS tree typically contains hundreds of thousands of Gaussians, naive long-term motion prediction is prohibitively expensive in memory and training data. Thus, an effective animation strategy is needed to reduce computational and data costs. Furthermore, since the training data are synthetic while testing targets real reconstructed trees, a robust 3D representation is essential to bridge the synthetic-to-real gap.

Our framework addresses these challenges with a two-stage pipeline. We first generate mesh motion and then bind Gaussians to the deforming mesh, allowing full 3DGS deformation while only modeling mesh dynamics Waczyńska et al. (2024); Gao et al. (2024). To further improve efficiency and generalization, we introduce a sparse voxel-based motion representation that both reduces the complexity of dense vertex deformations and mitigates the synthetic-to-real gap by converting irregular mesh sampling into a unified voxel structure. Moreover, inspired by previous work Li et al. (2024), we further model the motion of each voxel as a spectrum, which can model a long-term mesh motion using a few frequency components, further reducing the complexity for long-term motion generation. At last, we can treat the predicted 3D spectrum as 3D modal bases Li et al. (2024) for modal analysis and approximate 3D interaction by a summation of base motions. By doing so, it reduces the interaction simulation to about $18ms$, making it significantly faster than MPM-based simulation and enabling real-time interaction.

We evaluate our method through comparative experiments on various real-world scenes, showing that our approach produces more natural animations of trees swaying in the wind and dynamically interacting. We summarize our contributions as follows:

- We introduce *DynamicTree*, a novel framework for long-term motion generation of real-world trees, enabling realistic swaying animations.

- We propose a novel **sparse voxel spectrum** motion representation for efficient and long-term 4D generation. With the generated 3D spectrums, we can further perform fast simulation of interactive dynamics under external forces.

- To facilitate the generation of complex 3D tree motion, we contribute *4DTree*, a large-scale synthetic 4D tree dataset containing 8,786 animated tree meshes, each with 100-frame motion sequences.

## 2 RELATED WORK

### 2.1 TREE ANIMATION

Traditional methods for realistic tree animation typically involve creating a physical tree model and simulating its dynamics. For instance, Quigley et al. (2017) represents trees as collections of articulated rigid bodies connected by rotational springs, allowing for flexible yet physically plausible deformations. In contrast, Akagi (2012) introduces a particle-based approach where a link structure between particles replaces the traditional 3D tree model, enabling efficient computation of interaction forces and dynamic responses. Windy-Tree Pirk et al. (2014) incorporates a growth model along with Navier–Stokes equations and the Smoothed Particle Hydrodynamics method. Furthermore, Zhao & Barbič (2013) introduces a semi-automatic pipeline for interactive wind and drag simulations, which, however, requires heavy manual effort to make scanned trees simulation-ready and extensive artistic refinement of materials and lighting during rendering. In summary, although these prior methods can generate realistic dynamics, they remain limited to synthetic physical tree models or suffer from suboptimal rendering, leading to a noticeable gap in visual realism compared with 3DGS-based tree animations.

### 2.2 4D GENERATION

Recently, 4D content generation has gained growing attention in generative AI. These methods typically construct a static 3D model and then optimize its motion over time. Methods such as MAV3D Singer et al. (2023), Dream-in-4D Zheng et al. (2024), and CT4D Wu et al. (2024) use SDS to generate 3D models from text input, followed by video SDS to animate them. Comp4D leverages large language models (LLMs) Achiam et al. (2023) for motion generation, while 4Dynamic Yuan et al. (2024) bypasses optimization by using generated videos directly. Beyond text-conditioned, Animate124 Zhao et al. (2023) and DreamGaussian4D Ren et al. (2023) combine image-to-3D and video diffusion priors to optimize 4D models. EG4D Sun et al. (2024) and 4DGen Yin et al. (2023) first generate a dynamic video from the input image, and then use multi-view diffusion models Voleti et al. (2024); Liu et al. (2023) to generate multi-view sequences for optimizing the 4D representation. Despite recent progress, these methods rely on 2D motion priors from VDMs due to the lack of real 3D motion data. These limitations often result in inferior temporal and spatial coherence, causing noticeable artifacts in the optimized 3DGS results. Moreover, these approaches depend on scene-specific optimization, entailing substantial computational overhead.

### 2.3 PHYSICS-BASED 3DGS SIMULATION

Physics-based dynamic generation methods use the differentiable MPM simulation framework to optimize the dynamics of 3DGS. The pioneering work PhysGaussian Xie et al. (2024) employs a customized MPM formulation that bridges Newtonian dynamics and 3D Gaussian kernels, enabling the simulation of various material behaviors. To reduce manual parameter tuning, recent works combine MPM with VDMs or LLMs to estimate physical properties. For instance, PhysDreamer Zhang et al. (2024) and Dreamphysics Huang et al. (2024) integrate motion priors from VDMs with MPM, enabling the learning of dynamic properties such as Young's modulus and Poisson's ratio. Phys-Flow Comas et al. (2024) initializes parameters via GPT-4 Achiam et al. (2023) and further optimizes them using optical flow guidance. However, as precisely setting individual parameters for each part is challenging, these methods often assume uniform material properties across the entire object to simplify optimization. This facilitates global motion coherence but suppresses local deformation details and reduces visual realism in tree animation. Additionally, the high computational cost of MPM-based simulation limits its use in real-time applications.

### 2.4 SPECTRUM-BASED MOTION REPRESENTATION

Quasi-periodic motions of plants and trees are well-suited for spectrum-based modeling. Prior work shows they can be modeled as a superposition of a few harmonic oscillators at different frequencies Chuang et al. (2005); Davis (2016); Diener et al. (2009). Generative-Dynamics Li et al. (2024) leverages this property by reconstructing long videos from a few generated frequency components. Moreover, Abe et al. Davis et al. (2015) demonstrate that spectral volumes can serve as image-space modal bases for plausible interactive simulation via modal analysis. Building on this idea, Generative-Dynamics and ModalNeRF Petitjean et al. (2023) apply similar principles to image-

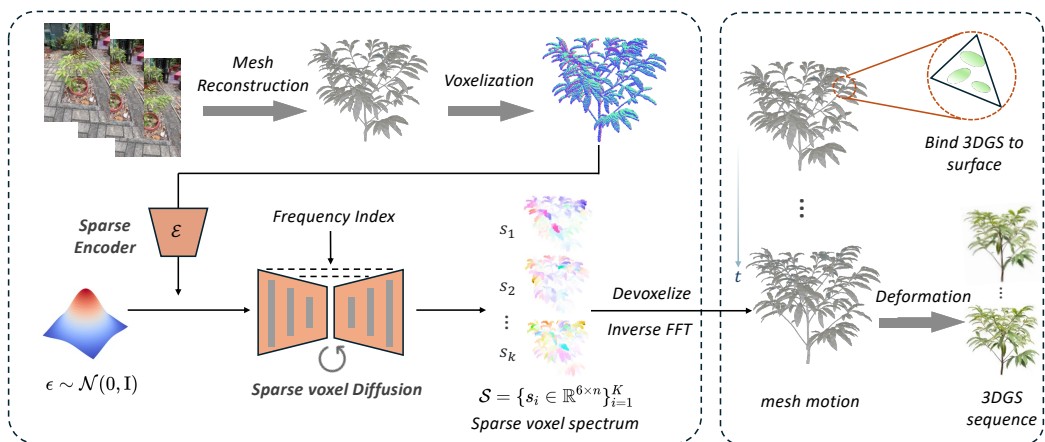

Figure 2: Our framework animates 3DGS trees in two stages: (1) spectrum-based motion generation in the frequency domain, and (2) deformation transfer to 3DGS through mesh binding. In the first stage, we extract the tree mesh from multi-view images, voxelize it, and encode it into a sparse voxel latent condition. A sparse voxel diffusion model then generates a compact motion representation $\mathcal{S}$, which is used to reconstruct mesh motion via devoxelization and inverse Fast Fourier Transform. In the second stage, 3DGS primitives are bound to the mesh surface and animated by its deformations.

space and implicit NeRF representation, achieving interactive dynamic simulations. Inspired by these works, we propose a sparse voxel spectrum representation that enables efficient long-term motion generation and interactive simulation for 3D trees.

## 3 METHODOLOGY

### 3.1 TASK FORMULATION

Given multi-view images of a static tree, our goal is to generate a 4D model as a deformed 3DGS sequence $\mathcal{G} = \{G_t \,|G_t = \{x_i^t, r_i^t, s_i^t, \sigma_i^t, c_i^t\}_{i=1}^H\}_{t=0}^T$, where $x_i^t, r_i^t, s_i^t, \sigma_i^t$ and $c_i^t$ denote the position, rotation, scale, opacity, and color of the $i$-th Gaussian primitive at frame $t$, respectively. This requires predicting temporal deformations of the static 3DGS: $\mathcal{D}_g = \{D_g^t | D_g^t = (\Delta x_i^t \in \mathbb{R}^3, \Delta r_i^t \in \mathbb{R}^4, \Delta s_i^t \in \mathbb{R}^3)\}_{i=1,t=1}^{H,T}$. Previous methods Yin et al. (2023); Comas et al. (2024) typically rely on optimization-based strategies to solve this problem, which are computationally expensive. We instead formulate this task as a conditional generation problem. To handle large-scale primitives efficiently, we propose a two-stage pipeline, named *DynamicTree*, as shown in Fig. 2. First, we introduce the *sparse voxel spectrum* (§3.2) representation to efficiently represent the motion. Then, we extract voxel grid conditions (§3.3) and employ a sparse voxel diffusion module (§3.4) to generate mesh motion. Subsequently, a two-stage optimization strategy is proposed in §3.5 to refine performance. Finally, we bind 3DGS on the animated mesh surface (§3.6) to compute $\mathcal{D}_g$.

### 3.2 SPARSE VOXEL SPECTRUM

The motion of a mesh sequence can be represented as $\mathcal{D}_m = \{D_m^t \in \mathbb{R}^{3 \times N} | t = 1, ..., T\}$, where $D_m^t(i)$ denotes the displacement vector of the $i$-th vertex relative to its initial frame position at time $t$. Although simpler than predicting the 3DGS deformation $\mathcal{D}_g = \{D_g^t \in \mathbb{R}^{10 \times H} | t = 1, ..., T\}$, where $D_g^t$ is the deformation of center, scale, and quaternion rotation for a Gaussian blob, mesh motion remains challenging due to the large number of vertices $N$ and frames $T$.

Prior works Wang et al. (2024); Lei et al. (2024) exploit the spatial sparsity of 3D motion using compact bases (e.g., Wang et al. (2024) drives 40k Gaussians with only 20 motion bases). In our case, tree-like motions also exhibit such sparsity. For example, vertices within the same leaf or local branch tend to show similar motion patterns. However, relying solely on sparse motion bases like Wang et al. (2024) will make it difficult to model fine-grained details due to the complexity of tree motion. Therefore, we propose representing tree motion using sparse voxels, where all vertices in a voxel share the same displacement. With this, to predict dense mesh motion, we only need to predict sparse voxel motion $\mathcal{D}_v = \{D_v^t \in \mathbb{R}^{3 \times n} | t = 1, ..., T\}$, where n is typically an order of magnitude smaller than $N$.

To ensure temporal consistency over long sequences, instead of autoregressive or time-conditioned generation Blattmann et al. (2023); Bertiche et al. (2023); Li et al. (2022), we draw inspiration from Generative-Dynamics Li et al. (2024), which models motion via low-frequency components of spectral volumes Davis et al. (2015). Inspired by this, we introduce the *sparse voxel spectrums* to represent 3D motion. Specifically, for sparse voxel motion $\mathcal{D}_v \in \mathbb{R}^{3 \times n \times T}$, we apply the Fast Fourier Transform (FFT) along the temporal dimension, resulting in a complex-valued frequency-domain representation $\hat{\mathcal{D}}_v \in \mathbb{C}^{3 \times n \times T}$, where each spatial displacement is decomposed into its corresponding components. Tree-like quasi-periodic motions are predominantly captured by the first $K$ frequency components. Thus, a compact representation $\hat{\mathcal{D}}_v^{(K)} \in \mathbb{C}^{3 \times n \times K}$ is sufficient for nearly lossless reconstruction of the full spectrum $\hat{\mathcal{D}}_v \in \mathbb{C}^{3 \times n \times T}$, with $K = 16$ following Li et al. (2024). Then, to facilitate the generation of these top-$K$ frequency components for the sparse voxel motion, we introduce the sparse voxel spectrum representation $\mathcal{S} = \{s_i \in \mathbb{R}^{6 \times n} | i = 1, ..., K\}$, where the dimension of size 6 corresponds to the real and imaginary parts of the $x$, $y$, and $z$ dimensions. Given this representation, we can reconstruct the mesh motion through the following operation:

$$\mathcal{D}_m = \text{Dev}(\text{iFFT}(\mathcal{S})) \tag{1}$$

where iFFT denotes the inverse FFT along the temporal dimension, and $\text{Dev}(\cdot)$ represents the de-voxelization process that maps sparse voxel displacements to dense mesh vertex motion.

### 3.3 VOXEL GRID CONDITION

To reduce the synthetic-to-real gap when using multi-view images as input, we condition the motion generation model on voxel grids. Given multi-view images of a static tree, we first reconstruct its mesh $M = (V, F)$ using an off-the-shelf method Guédon & Lepetit (2024), where $V = \{v_i \in \mathbb{R}^3\}_{i=1}^N$ and $F = \{f_j \subset \{1, \ldots, N\}\}_{j=1}^P$ denote the sets of vertices and faces, respectively. We then voxelize the mesh to obtain a sparse voxel grid $G$, which serves as the conditioning input.

### 3.4 SPARSE VOXEL DIFFUSION

Before performing the diffusion generation, the sparse voxel grid $G$ is encoded via a sparse encoder with several sparse convolutional blocks Williams et al. (2024), resulting in a compact latent representation $g \in \mathbb{R}^{d \times n}$ as geometric conditioning. Our sparse voxel diffusion module builds on the U-Net architecture introduced by XCube Ren et al. (2024). Specifically, to generate the sparse voxel spectrum $\mathcal{S} = \{s_i \in \mathbb{R}^{6 \times n} \mid i = 1, \ldots, K\}$ of the mesh motion, we condition the diffusion generation process on both the frequency index and the latent feature $g$, generating each frequency component separately. The diffusion process Ho et al. (2020) starts from pure Gaussian noise and iteratively predicts noise over $L$ Markov steps. At each step, we concatenate the latent feature $g$ with the noisy latent $s_l$, and inject the frequency embedding into every ResBlock of the sparse voxel U-Net through scale and shift operations.

### 3.5 OPTIMIZATION

We directly supervise the sparse voxel spectrums during training. To achieve this, we construct a 4D dataset of tree mesh motion sequences, which is detailed in Sec. 5. Given the constructed mesh motion, we first apply the FFT to obtain the spectrum for each vertex. Then, we voxelize the mesh motion spectrum to create the ground truth of sparse voxel spectrums.

During training, we diffuse the sparse voxel spectrum at each frequency component over $L$ diffusion steps and supervise the model's prediction using the following diffusion loss:

$$\mathcal{L}_{DM} = \mathbb{E}_{\epsilon \sim \mathcal{N}(0,I), l \sim \mathcal{U}(\{1,...,L\})} \left[ \|\epsilon - \epsilon_\theta(\mathbf{s}_l; l, g, f)\|^2 \right], \tag{2}$$

where $g$ and $f$ denote the sparse voxel latent condition and frequency embedding, respectively. Further, we find that using only the diffusion loss $\mathcal{L}_{DM}$ may lead to unrealistic motion, such as divergence of some branches, as shown in Fig. 5, because the problem is under-constrained. To address this, we introduce a Local Spectrum Smoothness (LSS) loss that encourages local consistency in the frequency domain, inspired by the physical prior proposed by Sorkine & Alexa (2007) that neighboring points tend to move similarly. Specifically, we compute discrepancies in the frequency-domain parameters between each point and its neighbors, weighted by spatial proximity:

$$\mathcal{L}_{\text{LSS}} = \frac{1}{N} \sum_{i=1}^{N} \sum_{j \in \mathcal{N}(i)} e^{-\alpha d_{ij}} \left( \|\text{Re}_i - \text{Re}_j\| + \lambda \|\text{Im}_i - \text{Im}_j\| \right),$$

where $\text{Re}_i$ and $\text{Im}_i$ denote the real and imaginary components of the spectrum at point $i$, $\mathcal{N}(i)$ represents its $\kappa$-nearest neighbors, $d_{ij}$ denotes the Euclidean distance between points $i$ and $j$, and $\lambda$ controls the weight of the imaginary part.

Moreover, we observe that naively combining both losses $\mathcal{L}_{DM}$ and $\mathcal{L}_{LSS}$ from the beginning of training would also lead to unstable learning. To address this issue, we adopt a two-stage training strategy: in the first stage, we train the model using only $\mathcal{L}_{DM}$ for a certain number of iterations; in the second stage, we introduce $\mathcal{L}_{LSS}$ to refine the spectral representation. This strategy significantly improves training stability and overall performance.

### 3.6 MESH-DRIVEN 3DGS ANIMATION

With the generated sparse voxel spectrums of a given mesh $M$, we need to decode it to a full motion field of 3DGS. To do that, we first devoxelize the sparse spectrum by assigning the same spectrum to all vertices within the same voxel. Then, we convert spectrums to the time-domain mesh motion $\mathcal{D}_m$ through the inverse Fast Fourier Transform.

With the recovered mesh motion, we then animate the 3DGS model, by binding Gaussian primitives to the mesh surface, as proposed by GaMeS Waczyńska et al. (2024). This operation can be viewed as a reparameterization: for each face $f_j = \{v_1, v_2, v_3\} \in \mathbb{R}^3$, we parameterize the attributes of its associated Gaussian primitive ($u$, $r$, and $s$) using the positions of the three vertices:

$$\begin{cases} \mu = \alpha_1 V_1 + \alpha_2 V_2 + \alpha_3 V_3, \\ r = [r_1(f_i), \ r_2(f_i), \ r_3(f_i)], \\ s = \text{diag}(s_1(f_i), \ s_2(f_i), \ s_3(f_i)), \end{cases} \tag{3}$$

where $\alpha_1$, $\alpha_2$, and $\alpha_3$ are learnable parameters, and $r_1$, $r_2$, $r_3$, $s_1$, $s_2$, and $s_3$ are parameterization functions. For details, please refer to Waczyńska et al. (2024). Through this binding strategy, we can compute the 3DGS deformation $\mathcal{D}_g$ directly from the mesh motion $\mathcal{D}_m$. This allows us to obtain the final deformed 3DGS sequence $\mathcal{G}$.

## 4 INTERACTIVE SIMULATION WITH MODAL ANALYSIS

Modal analysis is a technique used to decompose complex deformable motions into a set of fundamental vibration modes, each associated with a specific natural frequency. This approach is particularly well-suited for modeling the motion of systems composed of superpositions of harmonic oscillators, such as tree motion Diener et al. (2009); Habel et al. (2009). Given an external force $\mathbf{f}(t)$, we model all vertices of the tree mesh as a interconnected mass-spring-damper system $P$ to simulate the response $\mathcal{D}(t) = \{d_i(t) \in \mathbb{R}^3 \mid i \in P\}$. With this, we can construct the following equation of motion Shabana (1991):

$$M\ddot{d}(t) + C\dot{\mathbf{d}}(t) + K\mathbf{d}(t) = \mathbf{f}(t), \tag{4}$$

where $M$, $C$, and $K$ are the mass, damping, and stiffness matrices, respectively.

To solve this equation, we project it into modal space, resulting in $|P|$ independent equations Davis et al. (2015); Li et al. (2024):

$$m_i\ddot{q}_i(t) + c_i\dot{q}_i(t) + k_iq_i(t) = f_i(t), \tag{5}$$

where $m_i$, $c_i$, and $k_i$ correspond to the diagonal elements of the respective matrices. This is a standard second-order differential equation, which can be solved using explicit Euler integration. To perform the integration, we need to specify the initial modal displacement $q_i(0)$ and velocity $\dot{q}_i(0)$. These settings, along with the selection of $M$, $C$, and $K$, are based on the configurations described in Petitjean et al. (2023).

By solving for the modal responses $q^k(t)$ at each natural frequency, we can reconstruct the physical-space response using the corresponding mode shapes:

$$\mathcal{D}(t) = \sum_{k=1}^{K} \phi_k \cdot q^k(t). \tag{6}$$

Thanks to prior work Davis et al. (2015); Li et al. (2024); Petitjean et al. (2023) that has shown the spectrums of particle motion trajectories can be treated as modal bases, we can use the mesh motion spectrums computed in Sec. 3 as the modal shapes $\phi$ to solve the above equation and obtain the interactive dynamic response $\mathcal{D}(t)$.

Figure 3: Comparison with 4DGen Yin et al. (2023). We visualize the middle frame of the generated sequence, where our method preserves better 3D structures. Space-time slices are shown, with vertical and horizontal axes representing time and the spatial profile along the brown line, respectively.

## 5 DATASET

To facilitate the learning of complex 3D tree dynamics, we introduce *4DTree*, a large-scale 4D tree dataset containing $8,786$ animated tree meshes. Each instance includes a 100-frame animation, along with semantic labels for leaves and trunks, which can further support downstream tasks such as semantic segmentation.

To create 4D tree data, a straightforward approach is to use commercial physics-based simulation software, but this is time-consuming and impractical for large-scale datasets. Instead, we adopt the method from Weber & Penn (1995), which models trees as hierarchical branching structures and simulates wind-induced motion by treating stems as elastic rods coupled through oscillators, implemented via the Sapling Tree Gen add-on in Blender. Still, this approach involves many sensitive parameters, which, if set improperly, can cause unstable oscillations or unrealistic shapes. To ensure quality and consistency of our dataset, we adopt a three-stage pipeline during production: parameter tuning, automatic validation with scripts, and final visual filtering by human reviewers. Through this process, we construct a clean and diverse 4D tree dataset with complex dynamics. For Detailed procedures and data samples, please refer to the Appendix.

## 6 EXPERIMENTS

**Implementation**. We train our model from scratch without pre-trained models, taking 3.5 days on 8 RTX 4090 GPUs with a batch size of 48. During the first 40,000 iterations, we train the model using only the $\mathcal{L}_{DM}$ loss. Then, we introduce the $\mathcal{L}_{LSS}$ loss and continue training for an additional 30,000 iterations. For the $\mathcal{L}_{LSS}$ loss, we use the 5 nearest neighbors of each point, and both $\alpha$ and $\lambda$ are set to 0.5. We set the resolution of the sparse voxel spectrum to $128^3$, with an input resolution of $512^3$ for the sparse voxel encoder. This results in voxel latent conditions of dimensionality $d = 128$ at the $128^3$ resolution. When binding 3DGS primitives, we assign five Gaussians per face. We use the AdamW optimizer with an initial learning rate of $1 \times 10^{-4}$, which is halved every 20,000 iterations. During inference, we employ DDIM Song et al. (2020) with 100 sampling steps.

**Evaluation metrics.** To evaluate our method in real-world scenarios, we collect a test set of 13 real-world trees. For evaluation metrics, we follow prior works Yin et al. (2023); Chen et al. (2024) and use CLIP ViT-B/32 Radford et al. (2021) to measure both visual realism and temporal coherence. Specifically, we compute CLIP-I distance as the average CLIP distance between each frame and the input view, and CLIP-T distance as the average CLIP distance between consecutive frames. Furthermore, we conduct a user study on the rendered videos, focusing on four key aspects: motion authenticity (MA), motion complexity (MC), 3D structural consistency (3DSC), and visual quality (VQ). Below, we compare the results of motion generation and interaction simulation, respectively.

### 6.1 COMPARISON OF 3D ANIMATION

We select 4DGen Yin et al. (2023) as the baseline for motion generation. As shown in Fig. 3, the results of 4DGen often exhibit artifacts in fine details. Moreover, when the tree structures become more complex, 4DGen would fail to converge due to the degradation of motion generation in its un-

Figure 4: Interactive simulation comparison of different methods. We apply a dragging external force and then visualize the response of the scene, where our approach produces more natural oscillatory motion with finer-grained details. $t$ and $T$ denote the middle and final frames, respectively.

Table 1: Quantitative comparison of our method and other methods. The upper part is a comparison of 3D animation, and the lower part is a comparison of interactive simulation.

| Methods | CLIP Score | | User Study | | | | | Simulation time (ms/frame) |
|---|---|---|---|---|---|---|---|---|
| | CLIP-I↓ | CLIP-T↓ | MA↑ | MC↑ | SC↑ | VQ↑ | Overall↑ | |
| 4DGen | 0.0103 | 0.0094 | 4.6% | 8.5% | 2.1% | 2.1% | 4.3% | - |
| Ours | **0.0052** | **0.0021** | **95.4%** | **91.5%** | **97.9%** | **97.9%** | **95.7%** | - |
| PhysGaussian | 0.0061 | 0.0087 | 14.9% | 17.0% | **36.2%** | 12.8% | 20.2% | 1,800 |
| PhysFlow | 0.0047 | 0.0025 | 34.0% | 38.3% | 34.0% | 21.3% | 31.9% | 15,600 |
| Ours | **0.0038** | **0.0017** | **51.1%** | **44.7%** | 29.8% | **65.9%** | **47.9%** | **18.22** |

derlying VDM. For quantitative comparison, we evaluate only the input-view results to ensure a fair evaluation, as 4DGen performs poorly on novel views. The results in Table 1 demonstrate that our method achieves superior performance across all metrics. More animation results are reported in the Appendix. We strongly recommend viewing the supplementary videos for better visual assessment.

## 6.2 COMPARISON OF INTERACTIVE SIMULATION

For interactive dynamic simulation, we compare against state-of-the-art baselines PhysGaussian Xie et al. (2024) and PhysFlow Comas et al. (2024). As shown in Fig. 4, PhysGaussian and PhysFlow often produce unrealistic plastic deformations. Specifically, PhysGaussian deforms slowly with little rebound, while PhysFlow exhibits partial recovery but still lacks fine-grained elasticity at the branch and leaf level, producing overly global responses. In contrast, our method produces natural and elastic motions, with branches and leaves exhibiting distinct behaviors. Quantitative results from four viewpoints and simulation time are reported in Table 1. Our method not only outperforms the baselines on most metrics but also significantly reduces the simulation time. For each frame, our method takes only about 18 ms for simulation, with 13 ms for mesh motion computation via modal analysis, 2.57 ms for Gaussian deformations calculation, and 2.65 ms for rendering, achieving a real-time interaction. Note that PhysFlow requires additional parameter optimization, which results in significantly longer runtime. More simulation results can be found in the Appendix.

## 6.3 ABLATION STUDY

**The effect of training strategies**. We ablate the training strategy in Fig. 5. As shown, directly using $\mathcal{L}_D$ would often cause noticeable artifacts such as geometry scattering and divergence, while joint training with $\mathcal{L}_{LSS}$ alleviates but does not eliminate them. In contrast, our two-stage strategy first trains with $\mathcal{L}_D$ for several iterations before introducing $\mathcal{L}_{LSS}$, which effectively resolves these issues and greatly improves generalization.

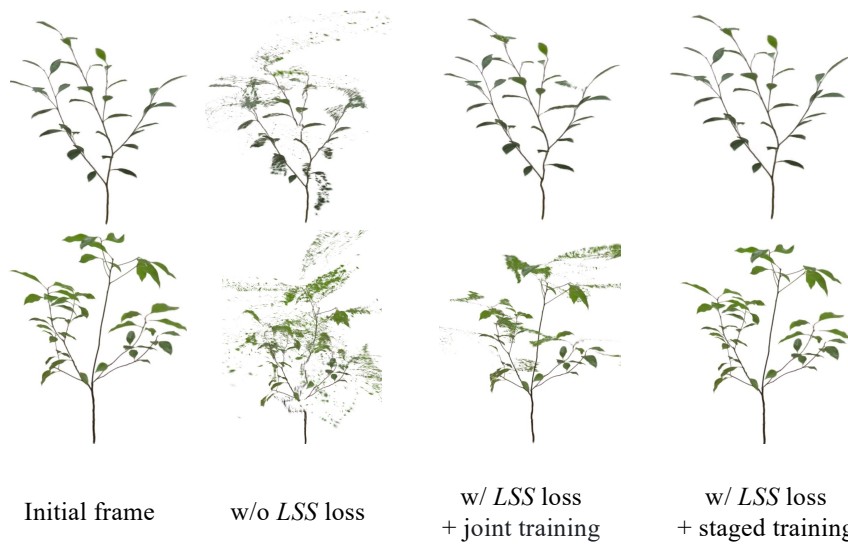

|  |  |  |  |
|:---:|:---:|:---:|:---:|
| Initial frame | w/o *LSS* loss | w/ *LSS* loss
+ joint training | w/ *LSS* loss
+ staged training |

Figure 5: Ablation of training strategies. Columns 2–4 show the middle frame of sequences generated by each strategy.

**The effect of different resolutions**. We compare the results of 3D animations at different sparse voxel spectrum resolutions in Table 2. These experiments are conducted on eight GeForce RTX 4090 GPUs, and due to memory constraints, we reduce the batch size as the resolution increases. As shown, the CLIP-I distance first decreases and then increases with increasing resolution. When the resolution exceeds 128, the improvement in CLIP-I becomes marginal, while the training cost continues to rise significantly. Therefore, we select 128 as our final resolution.

Moreover, we further analyze the synthetic-to-real gap through the performance degradation observed at a resolution of $512^3$. We find that at such a high resolution, the voxel grid becomes very fine and closely resembles point clouds, introducing a domain gap between training and inference. This is because real mesh vertices are generally noisier than synthetic ones. In contrast, using a resolution of $128^3$ partially mitigates this

Table 2: Ablation of different resolutions

| Resolution | Batch Size | Time | CLIP-I↓ |
|:---:|:---:|:---:|:---:|
| $32^3$ | 192 | 27h | 0.0097 |
| $64^3$ | 96 | 43h | 0.0069 |
| $128^3$ | 48 | 85h | 0.0039 |
| $256^3$ | 24 | 156h | 0.0037 |
| $512^3$ | 12 | 261h | 0.0056 |

issue, as multiple noisy points within the same voxel share the same motion pattern, leading to spatial smoothing that helps bridge the domain gap.

## 7 LIMITATION AND CONCLUSION

In this paper, we present *DynamicTree*, a novel framework for animating 3DGS trees. By introducing the sparse voxel spectrum representation, our method enables efficient long-term motion generation and real-time dynamic response to external forces. Furthermore, we also introduce a large-scale synthetic 4D tree dataset to support learning-based tree motion generation. Experimental results demonstrate that our approach achieves high-quality tree motion with strong temporal coherence and physical plausibility.

Although our method generates realistic 3D motion for real trees, several limitations remain. First, modal analysis is inherently a global linear approximation that shares vibration patterns across the entire object, potentially leading to synchronized motion between spatially distant regions. Second, mesh-driven 3DGS deformation may sometimes introduce artifacts in large deformation areas, which can be alleviated by increasing the number of Gaussians bound to faces in those regions. Finally, our experiments mainly target common immersive motions such as swaying, so the current dataset contains few large-scale deformations. In future work, we plan to augment the dataset with more large-deformation motions to address this limitation.

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

# A APPENDIX

## A.1 DATASET

As discussed in the paper, we implement the method of Weber & Penn (1995) as a plugin in Blender. However, generating a large-scale 4D tree dataset using this method remains challenging, as it involves numerous parameters and random sampling often produces invalid or unrealistic 4D trees. To ensure data quality and consistency, we adopt a three-stage pipeline to construct our dataset:

1. **Parameter Tuning**: Trees are controlled by many shape parameters. Fully random sampling over all of them tends to generate irregular or unrealistic trees, which can harm network training. Instead, we manually select key parameters such as branch count, height, branching angle, leaf count, etc., for stochastic variation. Other parameters are kept within small perturbation ranges. This approach ensures diversity while avoiding extreme or implausible deformations.

2. **Automatic Filtering**: After generating approximately 10,000 trees using the above strategy, we observe that some samples exhibit undesirable high-frequency oscillations, such as rapid back-and-forth motion at the root or excessive shaking in small branches. To filter out these cases, we apply the Fast Fourier Transform to each motion sequence and remove samples where the high-frequency components exceed a threshold.

3. **Manual Curation**: Finally, we perform visual inspection to eliminate edge cases such as unnatural branch clustering or physically implausible motion patterns.

Through this process, we curate a final set of 8,786 4D trees, with selected examples visualized in Fig. 6. For each tree, we first apply the FFT to its motion and then voxelize it. The spectrum of vertices within the same voxel is averaged to produce the final sparse voxel spectrum representation.

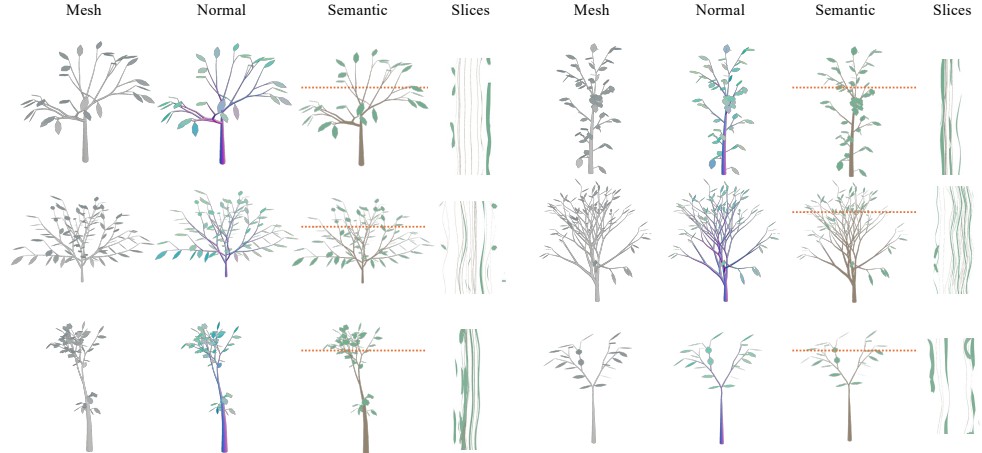

Figure 6: Examples from our synthetic dataset. To demonstrate the semantic labels, we render the leaves and trunk with two simple material settings. Users can replace these with more realistic materials for enhanced visual quality.

## A.2 NETWORK

The sparse encoder and sparse voxel diffusion U-Net are adapted from the basic modules proposed in XCube to better fit our conditioning input and spectral output. We report key parameter settings of these two components in Table 3

Table 3: Architecture Parameters

| Parameter | Sparse Encoder | Voxel Diffusion |
|---|---|---|
| Base channels | 32 | 128 |
| Depth | 3 | 2 |
| Channels multiple | - | [1, 2, 4, 4] |
| Head | - | 8 |
| Attention Resolution | - | [4,8] |

## A.3 VISUALIZATION RESULTS

Further visualizations and analytical details of the 3D animations and interactive dynamic simulations are presented in Fig. 8 and Fig. 7. However, to facilitate a comprehensive perceptual and qualitative evaluation of our method, we strongly recommend reviewing our supplementary videos.

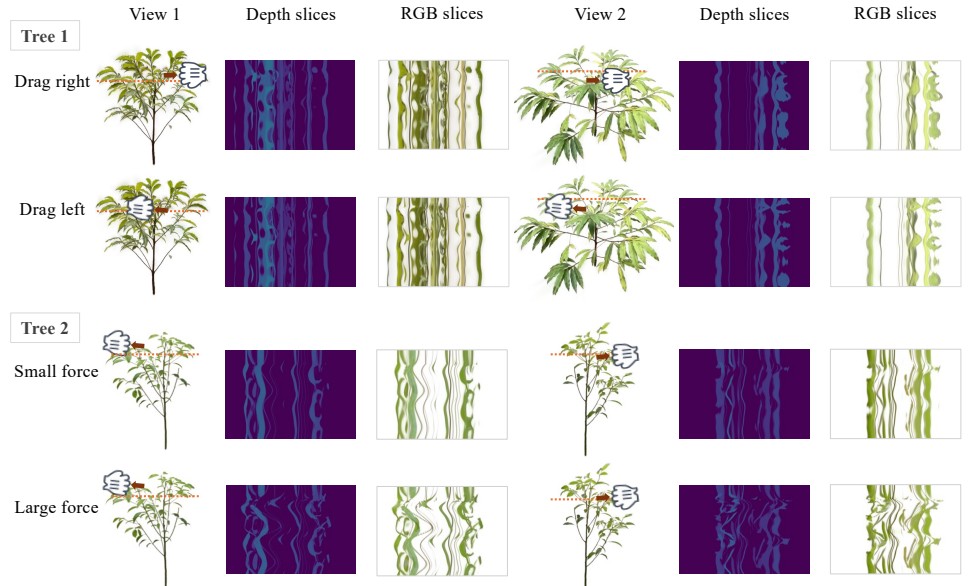

Figure 7: More results of interactive dynamic simulation. Our method can support interactive simulations involving forces with varying magnitudes and directions.

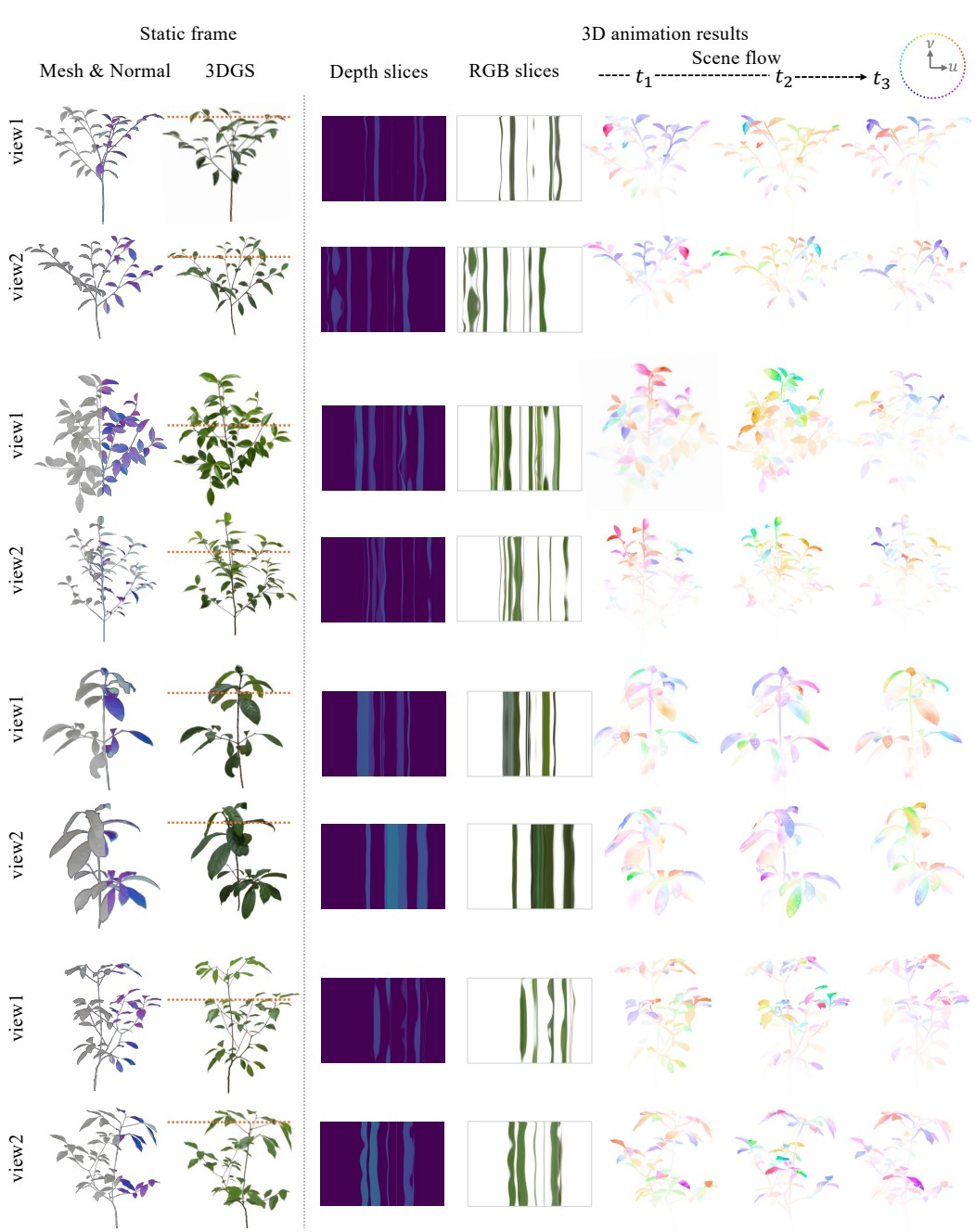

Figure 8: More results of 3D animation. For each scene, we visualize the space-time slices of depth and RGB videos from two viewpoints. We also show the scene flow of three mesh point cloud frames ($t_1 = 30, t_2 = 50, t_3 = 80$) in the generated sequence, with color coding following the strategy used in Hur & Roth (2020), where the movements $(u, v)$ along the $x$ and $z$ directions are encoded using standard optical flow coloring.

