# OpenReview forum: "DynamicTree: Interactive Real Tree Animation via Sparse Voxel Spectrum"
_ICLR.cc/2026/Conference — ICLR 2026 Conference Withdrawn Submission_

### Official Review · Reviewer_QBgY · 2025-10-27

**Soundness:** 3
**Presentation:** 3
**Contribution:** 3
**Rating:** 6
**Confidence:** 4

**Summary:**

This paper introduces DynamicTree, a novel framework that generates realistic long-term animations and enables real-time interactions for 3D Gaussian Splatting trees through an efficient sparse voxel spectrum representation.

**Strengths:**

（1）From a structural perspective, the paper is well-organized, with clear methodological exposition and illustrative diagrams, demonstrating logical coherence.
（2）In terms of quality, the comprehensive experiments demonstrate that the proposed method surpasses baseline approaches in both visual quality and temporal consistency, while achieving significant speed improvements in interactive simulation.
（3）The sparse voxel spectrum representation proposed in this paper presents an effective and innovative approach for compressing complex tree motions, combining spectral analysis with sparse voxel representation for 3DGS animation with groundbreaking originality.

**Weaknesses:**

（1）The evaluation primarily focuses on swaying motions, without exploring the performance under more complex conditions such as strong wind or heavy rain effects.
（2）While the method relies on synthetic training data with maximized realism in experiments, its generalization capability to diverse tree species in real-world scenarios requires further validation, and how to simplify the potential model deserves consideration.
（3）Although the proposed sparse voxel spectrum representation enhances efficiency, its reliance on synthetic training data constrains generalization capability. The model may exhibit performance degradation when encountering tree species with significantly different structures in real-world scenarios, and the applicability of the current method in computationally constrained environments remains insufficiently explored.

**Questions:**

（1）Can the sparse voxel spectrum representation support non-periodic motions, such as branch breakage or growth and decay sequences?
（2）Does the method require retraining when encountering tree species not covered in the training set? If so, please specify the required data and computational costs; if not, please provide theoretical basis for cross-species generalization.
（3）Could the current 128³ voxel resolution impose limitations on modeling motions of trees with intricate structures (e.g., willow branches)? Does there exist an adaptive resolution adjustment mechanism?
（4）Does the reported 18ms/frame performance include all preprocessing and post-processing stages? Please clarify the end-to-end actual latency in the complete pipeline, particularly regarding performance degradation when processing high-density trees.

---

### Official Review · Reviewer_QXqS · 2025-10-27

**Soundness:** 4
**Presentation:** 3
**Contribution:** 3
**Rating:** 6
**Confidence:** 4

**Summary:**

This manuscript presents DynamicTree, a framework that can synthesizes oscillatory motions and responses to external force for 3DGS trees reconstructed from multi-view images. Its core component is a diffusion model operating on sparse voxels that generates spectral volume for them. Given unstructured reconstructed meshes, per-vertex motion trajectories can be transformed from the queried spectrum in corresponding voxel. To train this model, they curate a tree mesh animation dataset using Blender and vowelize each vertex’s motion spectrum as ground truth. Following Generative Image Dynamics, the generated spectral volume also serves as a modal basis for simulating interactive dynamics. Experimental results show that the method can synthesizing natural, realistic motions. Compared with MPM-based simulation approaches, the proposed feed-forward method can achieve real-time interactive responses and without assumption on material properties. Compared with 4D generation methods for general objects, it demonstrates superiority on handling the fine-grained structures.

**Strengths:**

1.	The paper is well-organized and easy to read.
2.	The presented method is a solid and elegant extension of Generative Image Dynamics to 3D, yielding impressive results.
3.	By employs sparse voxel grids as a regular proxy, this work can robustly animate reconstructed irregular meshes and create a clean learning objective.

**Weaknesses:**

1.	Extending generative dynamics to 3D is an interesting exploration, but it should be noted that this also sacrifices the original scalability. Unlike 2D case where training data can be collected from existing videos in a relative lower cost, its 3D data is synthesized based on prior knowledge. It constrains scaling toward more diverse and larger-scale data--despite their commendable effort, the dataset remains limited in single category, and only learning the human designed patterns somewhat undermines its significance.
2.	There are some citation issues: (1) the reference to PhysFlow is incorrect—the cited Comas et al. (2024) is unrelated to this area; (2) when the author or publication is not part of the sentence, citations should appear in parentheses using \citep{}, it would be easier to read.
3.	The classification in L45–L49 is inaccurate. The listed works cannot be summarized as ”first create a static 3DGS model …… then use VDMs to optimize the 4D representation”.
4.	The implementation details of $L_{lss}$ needs clear description. Equation in L269 does not clarify how it relates to the network parameters in each training iteration. In the context of denoising diffusion training, what does the spectrum volume in this equation denote? Is it the original sample predicted at current step, or sampled by running full denoising process in each training iteration?
5.	It is suggested to provide more multi-view results (now only one multi-view video in the supp. video), which would be helpful to for reader to perceive the unique advantages of why extending generative dynamics to 3D.

**Questions:**

1.	L237 says the meshes are reconstructed via SUGAR, but this process should have already binded Gaussian to mesh. Why does L286 state that they are binded via GaMeS?
2.	The reconstructed tree is usually imperfect, how robust is the proposed method in such situation?

---

### Official Review · Reviewer_yccD · 2025-10-27

**Soundness:** 3
**Presentation:** 3
**Contribution:** 2
**Rating:** 4
**Confidence:** 3

**Summary:**

This work presents DynamicTree, a framework for generating realistic and interactive 3D tree animations from 3DGS reconstruction. It introduces a sparse voxel spectrum representation that models long-term 3D motion in the frequency domain, enabling efficient feed-forward animation and real-time response to external forces. The authors also contribute 4DTree, a large-scale synthetic dataset of animated trees for training and evaluation.

**Strengths:**

- Sparse voxel spectrum representation is introduced to bridge generative modeling and physical simulation for dynamic 3D tree animation, achieving both efficient long-horizon motion generation and interactive physical response compared to previous methods.

-  The manuscript is well written and clearly organized, with intuitive figures and videos that effectively communicate the two-stage pipeline, spectral representation, and qualitative results.

- A high-quality and relatively large-scale animated 3D tree dataset is presented to enhance reproducibility and future impact within the 3D/4D generation and world modeling communities.

**Weaknesses:**

1. **Limited problem scope**. The proposed method is designed specifically for tree animation and simulation, while most compared baselines target general dynamic objects or scenes. It remains unclear whether DynamicTree can generalize beyond trees, for example, to deformable or articulated objects such as cloth, humans, or other vegetation. Clarifying whether the proposed sparse voxel spectrum framework is object-agnostic or relies on tree-specific priors would strengthen the contribution’s generality.

2. **Insufficient baselines**. The comparison omits several recent and relevant MPM-based approaches, such as OmniPhysGS [1], PhysDreamer [2], and DreamPhysics [3], which also couple Gaussian Splatting with physics-guided dynamics. Including these baselines, or at least discussing why they are excluded, would provide a more comprehensive and convincing evaluation of performance and efficiency.

3. **Limited motion diversity**. The proposed dataset and method mainly include swaying motions with small deformations, lacking scenarios with large-amplitude or non-periodic dynamics (e.g., collisions, breaking, twisting). Consequently, the work does not demonstrate whether the method can handle strongly non-linear or non-harmonic motion, which is critical to validate the generality of the spectral representation.

[1] OmniPhysGS: 3D Constitutive Gaussians for General Physics-Based Dynamics Generation. ICLR 2025.

[2] Physdreamer: Physics-based Interaction with 3D Objects via Video Generation. ECCV 2024.

[3] DreamPhysics: Learning Physics-Based 3D Dynamics with Video Diffusion Priors. AAAI 2025.

**Questions:**

Please refer to the "Weaknesses" section.

---

### Official Review · Reviewer_CRA6 · 2025-10-29

**Soundness:** 2
**Presentation:** 3
**Contribution:** 2
**Rating:** 4
**Confidence:** 5

**Summary:**

This paper focuses on two problems: generating animation for reconstructed trees, and simulate their response to external forces. The technical idea is to learn the dynamics modes from a synthetic 4D tree dataset via a sparse voxel processing network, and then apply the modes during inference. Experiments show benefits over a few physics-based and learning-based baselines.

**Strengths:**

- A 4D tree dataset that looks realistic.
- Fast simulation via modal analysis.

**Weaknesses:**

My concerns are centered around the quality of the results.

- The animation does not seem to form a closed infinite loop. So there are some sudden changes after 3~4 seconds.
- The simulation does not look realistic to me. In particular, during 1:15 - 2:20 in the supplementary video, the force response always involves trivial vibrations from other tree branches than the one branch that is directly interacted with by the external force. Looking at 2:07-2:20, both small / large forces induce trivial vibrations on other branches.
- The baselines are somewhat outdated. The learning-based baseline is 4DGen which is published in late 2023, but apparently there are many recent 4D object generation works such as the ones (e.g., SV4D) from stability.ai. The physics-based baseline is PhysGaussian which is also outdated. Recent ones include PhysDreamer and quite some follow-ups.

Minor:
- The description on L150, "these methods often assume uniform material properties", is incorrect. The ones cited, PhysDreamer and DreamPhysics, learn spatially-varying materials, not uniform.

**Questions:**

Suggestions:
- Comparison to PhysDreamer (or other recent physics-based baselines) and SV4D (or other recent learning-based baselines).

---

### Official Review · Reviewer_F1qQ · 2025-10-29

**Soundness:** 2
**Presentation:** 2
**Contribution:** 2
**Rating:** 4
**Confidence:** 4

**Summary:**

This paper proposes a novel 4D generalization method named DynamicTREE, which is the first framework capable of generating long-term, interactive animations of 3D Gaussian Splatting trees. The authors introduce a sparse voxel spectrum motion representation for efficient and long-term 4D generation. They also contribute a new 4DTree dataset tailored for tree animation tasks.

**Strengths:**

1. The paper introduces a large-scale 4D tree dataset comprising thousands of animated tree meshes.
2. The authors propose a sparse voxel spectrum motion representation that supports efficient and long-term 4D generation.

**Weaknesses:**

1. The reviewer considers “Contribution 1” to be inadequately substantiated. The task of long-term motion generation is not new, and the framework appears to lack theoretical innovation. What specific problem does this framework address or mitigate?
2. The authors do not clarify whether the proposed 4DTree dataset will be publicly released. This raises doubts regarding the practical impact of Contribution 3.
3. While the authors propose a novel sparse voxel spectrum motion representation for efficient long-term 4D generation, they do not sufficiently demonstrate its effectiveness—for instance, by providing reconstruction results based on this representation. Moreover, how does the method perform on broader animation datasets such as ObjaverseXL-animation? Since tree animation is a subset of object animation, comparisons with feed-forward 4D generation methods (e.g., Gaussian Variational Field) should be included.

**Questions:**

Please refer to weakness.

---

### Note · Authors · 2025-11-13

I have read and agree with the venue's withdrawal policy on behalf of myself and my co-authors.